# A facile DNA coacervate platform for engineering wetting, engulfment, fusion and transient behavior
Wei Liu[1], Jie Deng[2], Siyu Song[1], Soumya Sethi[1] & Andreas Walther[1] ✉

Biomolecular coacervates are emerging models to understand biological systems and important building blocks for designer applications. DNA can be used to build up programmable coacervates, but often the processes and building blocks to make those are only available to specialists. Here, we report a simple approach for the formation of dynamic, multivalency-driven coacervates using long single-stranded DNA homopolymer in combination with a series of palindromic binders to serve as a synthetic coacervate droplet. We reveal details on how the length and sequence of the multivalent binders influence coacervate formation, how to introduce switching and autonomous behavior in reaction circuits, as well as how to engineer wetting, engulfment and fusion in multi-coacervate system. Our simple-to-use model DNA coacervates enhance the understanding of coacervate dynamics, fusion, phase transition mechanisms, and wetting behavior between coacervates, forming a solid foundation for the development of innovative synthetic and programmable coacervates for fundamental studies and applications.

Membraneless organelles (MLOs), also known as biomolecular condensates, form through liquid–liquid phase separation (LLPS) within cells and have been shown to play a crucial role in regulating biological functions[1–3]. Unlike traditional membrane-bound organelles, MLOs such as nucleoli, stress granules, and P-bodies function in the absence of a surrounding lipid bilayer. They are often formed by proteins or complexes of proteins and nucleic acids[4–6]. The unique properties of MLOs are predominantly driven by an array of weak interactions, including protein–protein, protein–RNA, electrostatic, and hydrophobic interactions, culminating in dynamic liquid properties[7–9]. These liquid characteristics influence the spatial organization, phase transition, and molecular interactions within cells, thereby regulating diverse cellular activities, including gene expression, cell adhesion, and cargo recruitment[10–12].

Inspired by natural LLPS-driven organelles, the fabrication of bioinspired LLPS systems, consisting of DNA, RNA, and proteins, has gained much ground in order to better understand biological systems using defined model systems, as well as for generating functional materials relevant to the molecular systems engineering world[13–17]. Despite the progress, many open questions remain, for instance, regarding the interactions between coacervates of different natures and of coacervates with soft interfaces. Interesting progress has also been made with respect to the uptake of coacervates

into liposomes or cells or, very recently, regarding LLPS in fibrillar environments[18–20].

DNA is an appealing building block to build up coacervates due to the programmable nature of interactions[21]. Various strategies have been employed to construct DNA coacervates and droplets, including the use of hybridization-mediated assembly of DNA nanostars or temperature-induced phase segregation and trapping of DNA[16,22–25]. These all-DNA artificial cell models can exhibit liquid-like behavior, beneficial to study adhesion, fission, fusion, and wetting[26–32]. For instance, Takinoue and co-workers investigated the impact of DNA sequences on the fusion dynamics of liquid-like droplets at elevated temperatures (above 43 °C)[28]. However, many of the approaches require specialist knowledge of DNA assembly techniques and well-designed annealing protocols, whereas simple mix-and-use protocols at one temperature would be very desirable.

In this study, we report an easy-to-use platform for all-DNA coacervates utilizing single-stranded DNA (ssDNA) components and multivalency-driven LLPS at physiological temperature (Fig. 1). We show that a series of palindromic domains can be flexibly hybridized to long ssDNA homopolymers composed solely of adenine nucleotides (polyA) to trigger coacervate formation, and to study dynamics as a function of the type of palindromic binder. Through the integration of DNA or RNA strands, we establish switchable systems for the controlled assembly and disassembly of

¹Life-Like Materials and Systems, Department of Chemistry, University of Mainz, Duesbergweg 10-14, 55128 Mainz, Germany. ²School of Chemistry and Chemical Engineering, Huazhong University of Science and Technology, Luoyu Road 1037, 430074 Wuhan, China. ✉e-mail: andreas.walther@uni-mainz.de

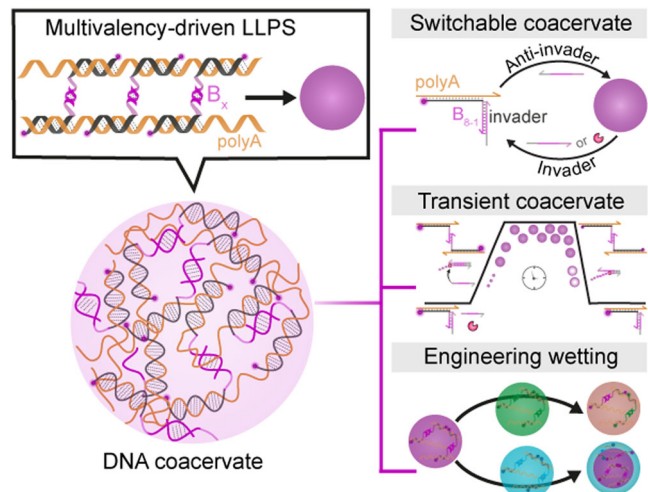

**Fig. 1 | Multivalency-driven all-DNA coacervates: coacervate preparation, switching, autonomous behavior, and engineered wetting for multiphase structures.** DNA coacervates are prepared via multivalency-driven liquid–liquid phase separation (LLPS) between a ssDNA polymer (polyA) and shorter self-complimentary binders ($B_x$). The introduction of DNA or RNA invaders triggers the formation of switchable and transient coacervates. Using two distinct coacervates with different palindromic domains allows controlling multiphase structures through engineering wetting.

coacervates. This advancement enables the realization of a transient, autonomous coacervate system[33]. Furthermore, the selective exploitation of the distinctive dynamic properties conferred by various palindromic domains allows to engineer multiphase structures through tailored wetting mechanisms. We anticipate that the facile formation mechanism and the flexible design of different dynamic behaviors of these DNA coacervates will pave the way for the engineering of functional DNA-based coacervates, with great potential to be followed up by non-experts.

## Results

### Formation of multivalency-driven coacervates

Our strategy for the fabrication of all-DNA coacervates employs a straightforward multivalency-driven LLPS approach based on the combination of a ssDNA homopolymer (polyA; A = adenine) and shorter binders ($B_x$) containing palindromic domains and a $T_{30}$ (T = thymine) domain for hybridization to the polyA backbone (Fig. 2a). The DNA homopolymer $A_{1500}$ was synthesized using terminal deoxynucleotidyl transferase (TdT)-catalyzed polymerization to feature 1500 repeating units[34,35]. Similar polyAs are, however, also available commercially. The binders, $B_x$, typically feature four components: (1) the complementary segment $T_{30}$ ensuring hybridization to the polymer $A_{1500}$ (melting temperature, $T_m$ ($A_{30}/T_{30}$) = 62.8 °C at 10 mM $Mg^{2+}$, 50 mM $Na^+$, by IDT), (2) an intermediate spacer providing both flexibility and a toehold ($S_7$), (3) the palindromic domain, $P_x$, facilitating multivalent interactions, and (4) a dye label. These structures are hence composed of $T_{30}–S_7–P_x$ (or shorter $B_x$), whereby $x$ signifies the length of the palindrome and thus the strength of the interaction. The strength of the palindrome also depends on the GC content. The table listing the $T_m$ in Fig. 2b gives a rough correlation of the binder strength.

We designed eight distinct binder strands ($T_{30}–S_7–P_x = B_x$), each characterized by a unique sequence and varying length of the palindromic domain (Fig. 2b). Through this approach, we aimed to understand the influence of these two factors (GC content and palindromic length) on the formation and dynamics of spherical coacervates. Coacervate systems were typically assembled by mixing $B_x$ with $A_{1500}$ in a molar ratio of 25:1 at 1.5 μM $T_{30}–S_7–P_x$ and 0.06 μM $A_{1500}$ at 37 °C in a buffer containing 10 mM $Mg^{2+}$ and 50 mM $K^+$ (pH = 7.9). Note that 0.06 μM $A_{1500}$ is equivalent to 3 μM $A_{30}$, and hence the $B_x$ segments can occupy 50% of all A repeat units in $A_{1500}$. Confocal laser scanning microscopy (CLSM) shows nicely spherical

coacervates for five of the eight systems (Fig. 2c, d; Supplementary Data 1). The system featuring the shortest $P_x$, comprised of merely four nucleotides (nt) and a low GC content ($P_{4-1}$), presents no distinguishable structure formation. Systems with the longest $P_x$, containing 10 or 12 nt ($P_{10}$, $P_{12}$), display excessive aggregation into non-spherical structures. This behavior roughly correlates with the $T_m$ of the individual palindromes, but the multivalent strengthening needs to be considered as well.

The absence of coacervate formation in the $A_{1500}/P_{4-1}$ system ($T_{m,P4-1} = 0$ °C) arises from the feeble multivalent binding due to the low GC content. Even though the other shorter $P_x$ systems ($x = 4-2, 6, 8-1, 8-2, 8-3$) have also calculated $T_m < 28$ °C, hence still significantly lower than the system temperature of 37 °C, they still lead to the formation of well-defined spherical coacervates due to efficient multivalency effects. This indicates a dynamic nature and binding/unbinding dynamics. In contrast, the aggregation in systems incorporating the longest $P_x$ strands ($x = 10, 12$) with $T_m = 38.7$ and $41.9$ °C can be attributed to the excessively rigid multivalent interactions without significant internal dynamics for rearrangement into spherical structures. Hence, for the fabrication of spherical all-DNA coacervate via multivalency-driven LLPS, the optimum length of $P_x$ containing four GC bases lies in the approximate range of 4–8 nt in total (Fig. 2c). This balances phase separation and sufficient rebinding dynamics on a multivalency level, yielding the desired spherical structural output.

We monitored a mixture of $A_{1500}$ and $B_{8-1}$ at 37 °C for visual tracking of the coacervate formation. After 0.5 h, a population of small coacervates with an average diameter of 0.7 μm appears. These small coacervates rapidly coalesce, gradually increase in size, and transform into larger entities with an average diameter of ca. 2.8 μm within 1 h. In the following 30 h, these structures continue to progressively coalesce and evolve into well-defined coacervates with ca. 8.8 μm diameter (Fig. 2e, Supplementary Fig. 1). This coalescence and growth further highlight the dynamic behavior within the phase-separated coacervate state. Fluorescence recovery after photobleaching (FRAP) measurements reveal a partial, limited recovery within 1.5 h post bleaching for the three coacervates with 8 nt in the palindrome. At the same time, partial fusion events can be clearly observed (Supplementary Fig. 2; arrows therein highlight fusion events; Supplementary Fig. 3 shows the minor effect of palindromic domains with different lengths = 4, 6, 8 nt). Hence, these coacervates are sufficiently dynamic to allow for fusion. Most interestingly, the coacervates formed by $B_{8-2}$ show the quickest recovery at the coacervate surface. Hence, despite the strong similarities in the different binders of the $B_{8-y}$ series, subtle differences in behavior can occur.

### DNA- and RNA-triggered coacervate dynamics: switchable and transient systems

In the above investigation, we discussed the mechanism and conditions guiding the formation of our multivalency-driven coacervates. To achieve switchable and transient coacervate systems, we employed tools from DNA and RNA nanoscience to coacervates formed with the binder $B_{8-1}$. For the establishment of switchable coacervates, we introduced a DNA invader consisting of a domain complementary to the $S_7–P_{8-1}$ segment of the binder, along with an additional dangling overhang acting as a toehold for subsequent reactions (Fig. 3a; all sequences in Table S1). The addition of this invader to already formed coacervates disengages the multivalent interactions via strand displacement of the $P_x/P_x$ palindrome. CLSM images demonstrate the disappearance of the coacervates upon the addition of the DNA invader in less than 10 minutes (Fig. 3b). Furthermore, the introduction of an additional DNA anti-invader, which can hybridize with the DNA invader at its dangling overhang, triggers displacement of the invader and the re-engagement of the palindromic hybridization site followed by reassembly of the coacervate. This process enables simple isothermal switching of the DNA coacervates.

This concept can be extended to an RNA invader capable of complementary interaction with the $S_7–P_{8-1}$ segment of the binder (Fig. 3c, d, Supplementary Fig. 4). When RNA invaders are introduced, the coacervates disassemble rapidly in less than 10 minutes. Coacervate reassembly occurs upon the addition of RNase H, which specifically degrades RNA hybridized

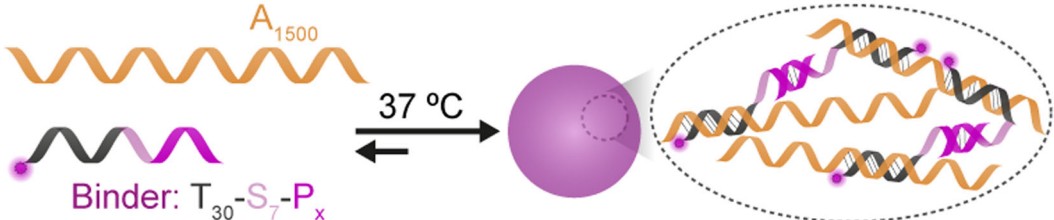

## a Coacervate formation via multivalency-driven LLPS

$A_{1500}$

37 °C

Binder: $T_{30}$-$S_7$-$P_x$

## b Palindromic domain

| $P_x$ | Sequence (5'→3') | $T_m$ (°C) |
|---|---|---|
| $P_{4\text{-}1}$ | GTAC | 0.0 |
| $P_{4\text{-}2}$ | GCGC | 5.6 |
| $P_6$ | GCTAGC | 17.3 |
| $P_{8\text{-}1}$ | GTAGCTAC | 27.6 |
| $P_{8\text{-}2}$ | GCTATAGC | 27.0 |
| $P_{8\text{-}3}$ | CAGTACTG | 26.5 |
| $P_{10}$ | AGCTATAGCT | 38.7 |
| $P_{12}$ | TAGCTATAGCTA | 41.9 |

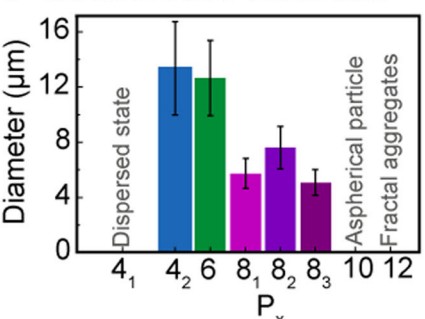

## c Coacervate diameter

## d Effect of palindromic domain length

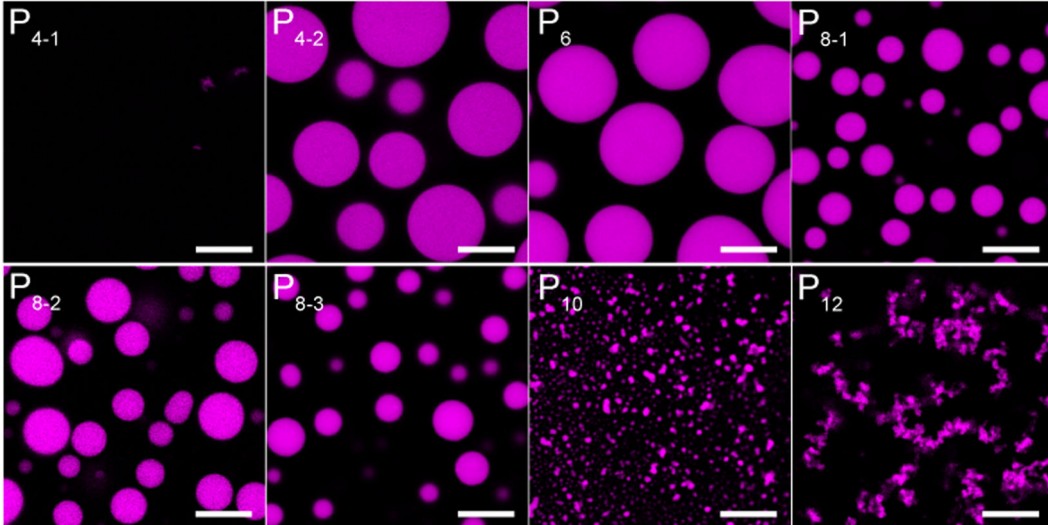

## e Coacervate formation over time for $A_{1500}$/$B_{8\text{-}1}$

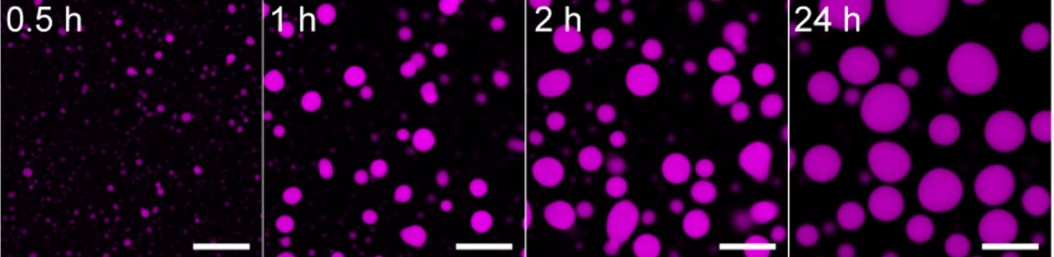

**Fig. 2 | Mechanism of the formation of multivalency-driven coacervates and the palindromic domain effect on the formation. a** Multivalency-driven coacervates by a long ssDNA ($A_{1500}$) mixed with short binder ssDNA consisting of a complementary strand $T_{30}$, a spacer $S_7$ and a palindromic multivalency domain $P_x$. **b** Sequences and melting temperatures ($T_m$) of the palindromic domains at 10 mM $Mg^{2+}$. **c** Size distribution of coacervates in systems containing 0.06 μM $A_{1500}$ and 1.5 μM binder at 37 °C after a 4 h reaction period with various palindromic domains. Error bars correspond to standard deviations of ca. 150 coacervates. **d** CLSM images of systems containing 0.06 μM $A_{1500}$ and 1.5 μM binder at 37 °C after a 4 h reaction period with different palindromic domains. **e** Time-dependent CLSM images of $A_{1500}$/$B_{8\text{-}1}$. The fluorescent labels are on the $B_x$. The experiments were conducted at 37 °C with gentle rotation at 80 rpm. Scale bars: 10 μm.

**Fig. 3 | DNA- and RNA-triggered switchable coacervates and RNA-regulated transient coacervates. a** and **b** Illustration and CLSM images of DNA-triggered switchable coacervate systems. 2.25 μM (1.5 eq.) of DNA invader was introduced to the coacervates formed from 0.06 μM $A_{1500}$ and 1.5 μM (1.0 eq.) of $B_{8-1}$, resulting in the disassembly of the coacervates. Subsequently, the addition of 3 μM (2.0 eq.) of DNA anti-invader caused the reformation of coacervates. **c** and **d** Illustration and CLSM images of RNA-triggered switching between coacervate and solution states. 4.5 μM (3 eq.) of RNA invader was introduced to the coacervates formed from 0.06 μM $A_{1500}$ and 1.5 μM (1.0 eq.) of $B_{8-1}$, resulting in the disassembly of the coacervates. Subsequently, the addition of 0.1 U μL$^{-1}$ of RNase H triggered the assembly of coacervates. **e** and **f** Illustration and CLSM images of RNA-triggered switchable coacervate of solution-to-coacervate-to-solution. 15 μM (10 eq.) of RNA anti-invader was introduced into the complex comprising 0.06 μM $A_{1500}$, 1.5 μM $B_{8-1}$, and 2.25 μM DNA invader, leading to the assembly of the coacervates. Subsequently, the addition of 0.1 U μL$^{-1}$ of RNase H caused the coacervates to disassemble into a homogeneous solution. A few tiny objects in the initial state may be due to the incomplete dissolution of the dye-bearing strand adsorbed in the microscopy chamber. **g–k** RNA-regulated transient coacervate disassembly with the introduction of RNA invader and RNase H: **g** Schematic illustration. **h** Time-dependent plot of the transient coacervates with tunable reassembly times by varying the concentration of RNA invaders. The green line indicates starting from a similar population of coacervates. Error bars are the standard deviation of ca. 20 droplet counts. **i** Controlled reassembly times via the concentration ratios of RNA invader/binder. Error bars correspond to standard deviations from duplicates. **j** Controlled reassembly times via the RNase H concentrations. Error bars correspond to standard deviations from duplicates. **k** Time-dependent CLSM images of transient coacervates with 15 μM (10 eq.) RNA invader and 0.1 U μL$^{-1}$ RNase H. **l** and **m** RNA-regulated transient coacervate assembly with the introduction of RNA anti-invader and RNase H: **l** Schematic illustration. **m** Time-dependent CLSM images of transient coacervates with 45 μM (30 eq.) RNA anti-invader and 0.1 U μL$^{-1}$ RNase H. The experiments were conducted at 37 °C with gentle rotation at 80 rpm. Scale bars: 10 μm.

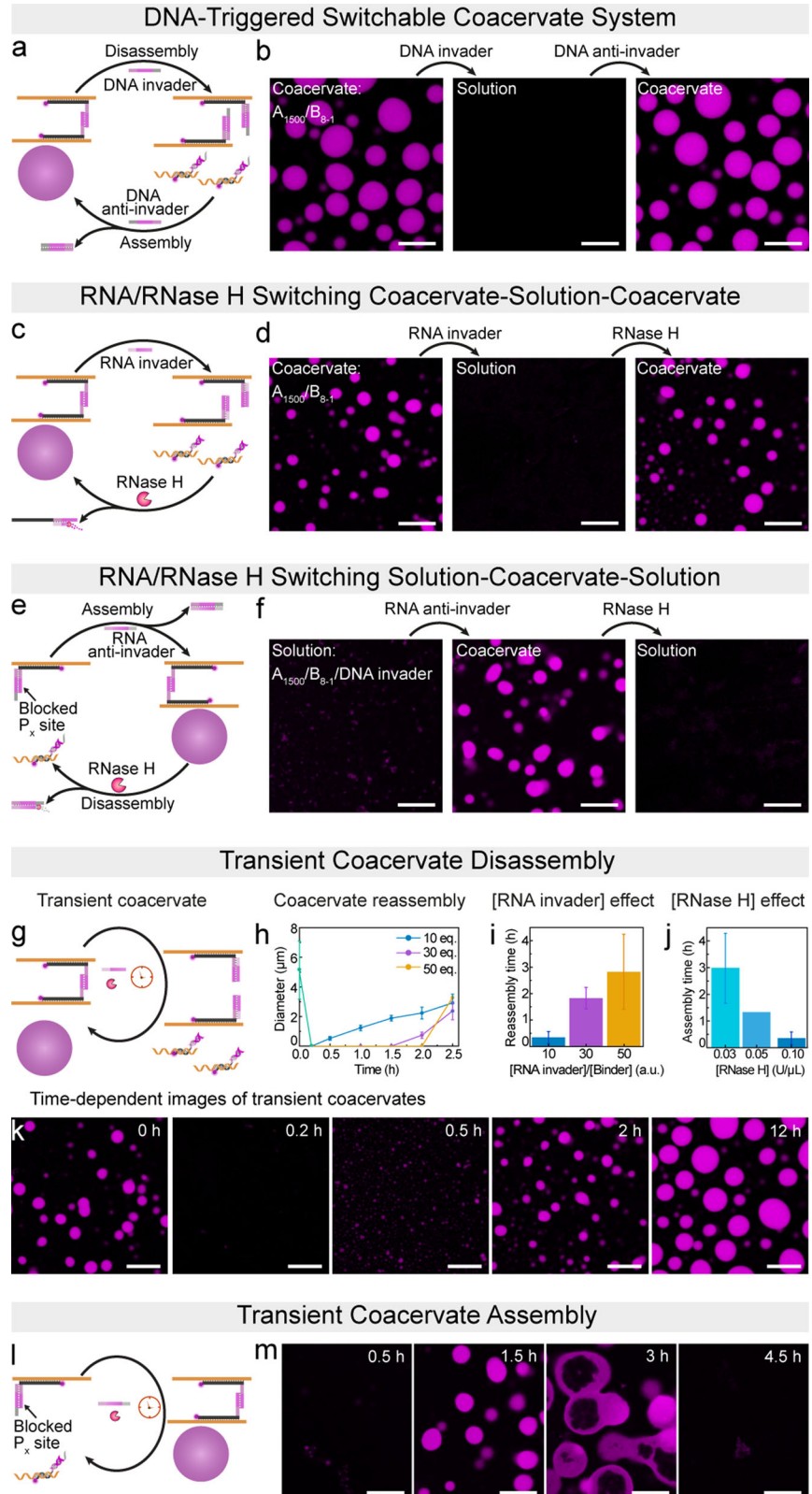

with DNA. A toehold is no longer needed as for the DNA invader. The use of an enzyme allows, in principle, to fine-tune the kinetics of the re-engagement by simply changing its concentration. Furthermore, the switching process can be inverted by introducing an RNA anti-invader that removes a blocker strand from the $B_x$ domain to trigger assembly. Subsequent addition of RNase H thereafter affords disassembly of the

coacervate systems by degradation of the RNA anti-invader and subsequent reblocking of the $B_x$ with the newly liberated original ssDNA block strand (Fig. 3e,f).

Building upon the RNA-triggered switchable coacervate systems, we asked whether autonomous and transient coacervate dissolution, as well as coacervate formation, would be possible by the addition of RNA trigger

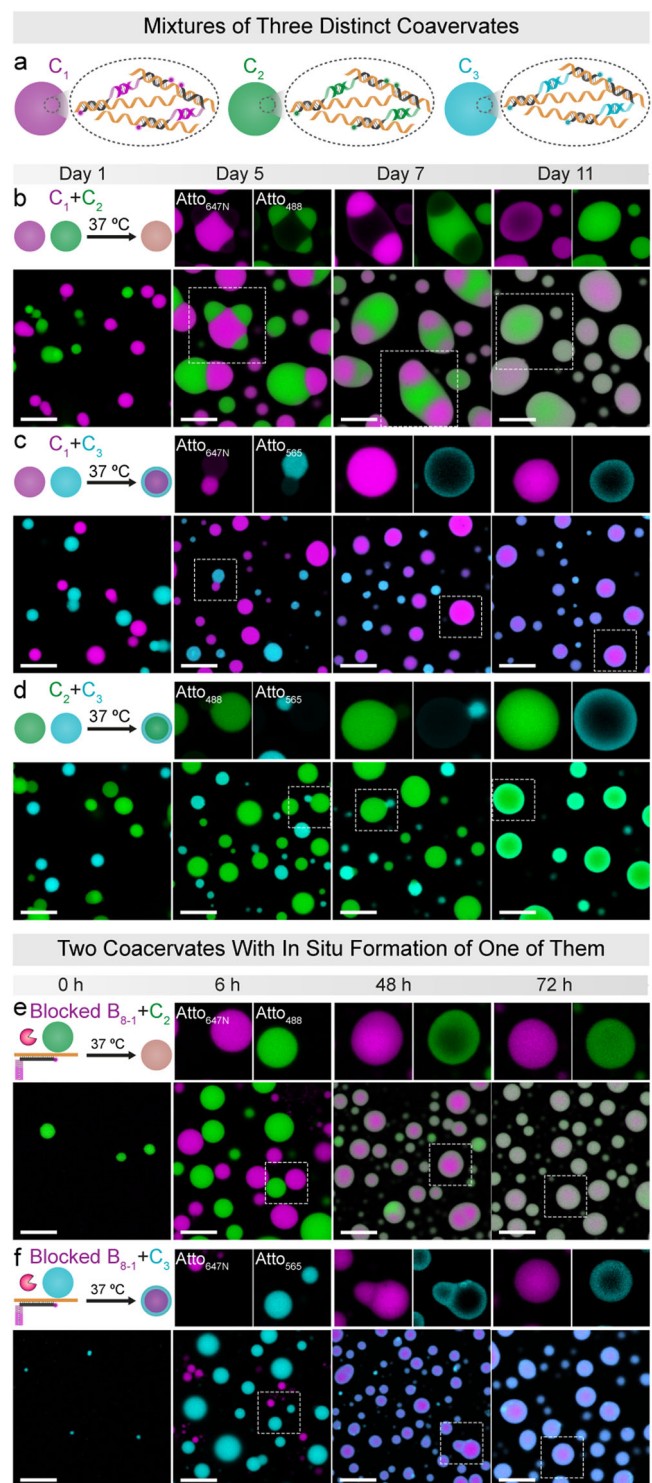

**Fig. 4 | Engineering wetting between coacervates formed with diverse palindromic sequences at 37 °C.** The coacervates were prepared using a mixture containing a final concentration of 0.06 μM of $A_{1500}$ and 1.5 μM of the respective $B_x$. **a** Illustration of three distinct coacervate structures ($C_1$ in magenta, $C_2$ in green, and $C_3$ in cyan) synthesized from different binders ($Atto_{647N}-B_{8-1}$, $Atto_{488}-B_{8-2}$, and $Atto_{565}-B_{8-3}$) with various palindromic sequences ($P_{8-1}$, $P_{8-2}$, and $P_{8-3}$, as showed in Fig. 2b, c). The experiments were performed at 37 °C with gentle rotation at 80 rpm unless otherwise specified. **b** Wetting behavior between distinct $C_1$ and $C_2$, leading to the formation of mixed coacervates. **c** Wetting behavior between $C_1$ and $C_3$, resulting in the formation of core–shell structures. **d** Wetting behavior between $C_2$ and $C_3$, contributing to the formation of core–shell structures. The experiments were carried out at 37 °C without rotation to minimize collisions. **e** Behavior during RNase H-initiated, slow coacervation of $C_1$ in a dispersion of preformed $C_2$. 0.1 U μL$^{-1}$ RNase H and preformed $C_2$ were introduced into the complex comprising 0.06 μM $A_{1500}$, 1.5 μM of $B_{8-1}$, and 2.25 μM RNA invader. **f** Behavior during RNase H-initiated, slow coacervation of $C_1$ to preformed $C_3$. 0.1 U μL$^{-1}$ RNase H and preformed $C_3$ were introduced into the complex comprising 0.06 μM $A_{1500}$, 1.5 μM of $B_{8-1}$, and 2.25 μM RNA invader. The experiments were conducted at 37 °C with gentle rotation at 80 rpm. Scale bars: 10 μm.

are observed because of the gradual degradation of the RNA invader by the already present RNase H. As RNA degradation continues, the reformation of well-defined coacervates occurs (Fig. 3h, k; Supplementary Data 1). Importantly, the reassembly time (defined as the point where coacervates reappear in CLSM) of these transient coacervates can be adjusted within a range of 0.3–2.9 h by either augmenting the quantity of RNA invader or reducing the concentration of RNase H (Fig. 3i, j; Supplementary Figs. 5 and 6; Supplementary Data 1).

The opposite pathway, i.e., transient coacervates, can be accomplished by integrating the $A_{1500}$/$B_{8-1}$/DNA-invader complex with an excess of RNA anti-invader and a low concentration of RNase H (Fig. 3l). The RNA anti-invader quickly removes the DNA strand blocking the multivalent interaction, and thereby activated coacervation. By balancing the competition between anti-invader/invader hybridization and anti-invader degradation by RNase H and allowing the necessary time for coacervate formation, well-defined coacervate structures are observed after 1.5 h (Fig. 3m, Supplementary Fig. 7). After 3 h, these coacervates gradually disappear. Strikingly, hollow capsules appear, which indicates entrapment of the RNase H due to affinity to RNA (and potentially DNA) inside the coacervates and degradation from the inside. The relatively high stability of the vacuoles without diffusion to the surface likely originates from the viscoelastic character of the coacervates with limited dynamics (FRAP in Supplementary Fig. 2). Such vacuole formation has been observed earlier for enzymatic degradation of DNA nanostar condensates when adding a restriction enzyme from the outside and was associated with the dynamics of the condensate and enzyme migration dynamics[39]. Our mechanism is, however, different as the restriction enzyme is present from the start and entrapped and does not need to diffuse into the condensate. Both approaches highlight the development of RNA-mediated autonomous transient coacervate systems that function by simultaneously utilizing RNA strands and the endoribonuclease. These systems will be used below for slow activation of binary coacervate systems.

### Engineering wetting between coacervates

After understanding how the combination of $A_{1500}$ and $B_x$ can lead to the formation of dynamic and switchable coacervates, we hypothesized that these simple-to-prepare coacervates can be used to engineer coacervate mixtures and interactions using various binders. To this end, we prepared three distinct coacervates: $C_1$ in magenta, $C_2$ in green, and $C_3$ in cyan, using different binders $B_{8-1}$, $B_{8-2}$, and $B_{8-3}$, respectively, wherein each binder is associated with a specific and selectively binding palindrome sequence ($P_{8-1}$, $P_{8-2}$, and $P_{8-3}$, as shown in Figs. 2b, c and 4a). After the initial formation of the individual coacervates (time = 1 h; 37 °C), we combined two of them to understand whether they remain segregated or potentially fuse despite being built by different palindromic binders.

strands into RNase H-loaded systems[36–38]. Fig. 3g–k displays transient coacervate disassembly. Coacervates formed by $A_{1500}$ and $B_{8-1}$ were first generated and then combined with an excess of RNA invader (10 eq. of binder) at a low concentration of RNase H (0.1 U μL$^{-1}$). Before introducing the RNA invader, CLSM images show the initially formed coacervates (Fig. 3k). The subsequent simultaneous addition of RNA invader and RNase H set off an autonomous and dynamic transformation between coacervate and solution states. Within 10 min, the coacervates disassemble into a fully homogeneous solution due to rapid hybridization of $B_{8-1}$ and RNA invader, which blocks multivalent interactions. After 0.5 h, some minor coacervates

Following an 11-day incubation period at 37 °C, a rich diversity of morphologies emerges. In general, the process follows initial contact at early times, subsequent engulfment, formation of a preferred wetting layer, and ultimately complete mixing (Fig. 4). The harmonization process occurs on different time scales for the different coacervate mixtures and is not complete for mixtures of $C_1$ (magenta)/$C_3$ (cyan) and of $C_2$ (green)/$C_3$ (cyan), for which distinct core–shell morphologies persist with slow homogenization even after 11 days (Fig. 4c, d), These observations are in line with the diminished dynamics of $C_3$ relative to $C_1$ and $C_2$ (Supplementary Fig. 2). The tendency to form the wetting layer on the surface follows $C_1 < C_2 < C_3$. It leaves us to speculate that the small differences in palindrome binder—despite being very similar in structure and thermodynamics—leads to slightly different surface tensions. It is interesting to note that the most hydrophilic dye ($Atto_{488}$) located on $C_2$ (green) does not form the most effective surface wetting layer (Fig. 4d). Concerning the gradual homogenization, control experiments by gel electrophoresis point to a gradual strand displacement within simplified model system of $A_{37}$/$B_{8-1}$ and $B_{8-2}$ (Supplementary Fig. 8), wherein the initially bound $B_{8-1}$ strand is undergoing slow exchange with $B_{8-2}$. This process may aid in compatibilization. Note that NUPACK simulations do not indicate cross-talk/off-target binding between the different $B_{8-x}$ binders down to a temperature of 5 °C.

To further investigate the emergence of wetting phenomena, we employed the autonomous formation of one coacervate system in the presence of an already formed different coacervate. This connects to Fig. 3c, where RNase H-mediated digestion of RNA was used to trigger coacervation. When using slow RNase H-mediated formation of $C_1$ in the presence of already preformed $C_2$ (Fig. 4e), the data shows homogeneous nucleation of distinct magenta $C_1$ structures in the presence of preformed $C_2$. Heterogeneous nucleation is absent. Over time, these discrete coacervates form a unified coacervate. When slowly activating $C_1$ to preformed $C_3$ (Fig. 4f), again homogeneous nucleation takes place followed by the formation of core-shell structures. Hence the morphological development is independent of the nucleation pathway and the temporal orchestration. No matter whether preformed coacervates are mixed or whether one is grown slowly, similar fully mixed or core/shell structures appear. This points to the robustness of the process. Overall, this approach shows how to control diverse morphologies in the presence of two distinct coacervate systems, thereby not only achieving wetting control but also mimicking the formation processes and wetting behaviors of the biological organelles within cells[29,40–42].

Obviously, the wetting process between two preformed coacervates takes more than one week because of the slow mobility of the coacervates, as well as the very slow possible exchange of binder segments. To gain a better understanding of the wetting behavior and accelerate this process, we employed a so-called $C_{31}$ system together with the $C_1$ coacervates (Fig. 5). The $C_{31}$ coacervate was engineered to contain mixtures of its original $B_{8-3}$ binder and either 25 or 50 mol% of the $B_{8-1}$ binder (Fig. 5a). The latter serves to strengthen and accelerate interactions between the $C_{31}$ and the $C_1$.

Upon mixing $C_1$ and $C_{31}$ (Fig. 5c, d), core–shell structures visually identical to the $C_1$/$C_3$ system are formed (Figs. 4c and 5b), but with different formation times, based on the ratio of $B_{8-3}$ to $B_{8-1}$. $C_{31}$ with a $B_{8-3}$:$B_{8-1}$ ratio of 75%:25% leads to the observation of Janus-like structures within 1 day. Following this, multicompartment structures occur after 3 days, and the final core–shell structures emerge within 7 days (Fig. 5c). Furthermore, changing the $B_{8-3}$:$B_{8-1}$ ratio to 50%:50% significantly enhances the wetting behavior, resulting in the emergence of the final core–shell structures within 3 days, instead of 7 days (Fig. 5d). This phenomenon indicates that the wetting speed can be effectively modulated by tuning the multivalent interactions of the binders between two coacervates.

The wetting behavior of $C_1$ and $C_{31}$ helps to understand the interaction in a mixture of pure, distinct $C_1$ and $C_{31}$. In the mixture of $C_1$ and $C_3$, as mentioned above, the unbound free A repeat units on $A_{1500}$ within coacervates present opportunities for strand exchange between $B_{8-1}$ in the $C_1$ coacervate and $B_{8-3}$ in the $C_3$ coacervate (Fig. 5e; Supplementary Fig. 8). This strand displacement process leads to an increased presence of $B_{8-3}$ in $C_1$ as well as of $B_{8-1}$ in $C_3$ that enables multivalent interactions between the original $C_1$ and $C_3$ via wetting and engulfment. This strand displacement behavior can occur when the coacervates come into contact. Subsequently, the strand exchange enhances the emergence of complex structures through wetting, fusion, and engulfment. The process is sketched in Fig. 5e. We hypothesize that strand exchange may also occur very slowly, even when the coacervates are at a distance from each other through complete detachment of the binders and their diffusion (equilibrium dynamics of dsDNA). It is reasonable to assume that elongation of the $T_x$ segment in the binders would further limit the molecular exchange and thus slow down or even prevent fusion.

## Conclusions

In summary, we introduced a simple and versatile platform for programmable DNA coacervates. In contrast to many other DNA coacervate systems, this model system does not require much knowledge on the design of DNA nanostars, complicated synthesis or well-designed annealing protocols, but operates on a simple mix-and-use principle. Furthermore, due to the presence of a polymeric component (polyA), a different viscoelastic regime can be generated within the coacervates as compared to nanostar-based droplets, because polymer entanglements can be present for long polyA, as well as higher multivalency effect. FRAP measurements indicate a slow and partial recovery in the coacervates, contrasting with the rapid and almost complete recovery reported in nanostar-based droplets by previous studies[22,43–45]. Even though we synthesized the polyA building block in-house, it is commercially available, and preliminary experiments show that the commercial polyA also undergoes coacervation (Supplementary Fig. 9).

Using established strand displacement switching principles of DNA nanoscience, we demonstrated simple switching of our coacervates. The assembly/disassembly switching can also be achieved using RNA/RNase H switches that allow fine-tuning of the kinetics and the design of autonomous systems, where recruitment of the enzymes into the coacervate droplet was observed. The liquid-like properties and small differences in the palindrome binder of these coacervates paved the way for engineering wetting, clustering, engulfment, and merging of the distinct coacervate droplets. The interaction of these coacervates can be simply engineered by co-hybridization with other linkers, which should allow for sequential mixing processes in the future.

The facile tunability and the facile options for functionalization should enable fundamental studies into coacervate dynamics and structures, as well as open doors for applications in delivery, synthetic biology, and material science.

## Methods

### Synthesis of ssDNA polymer $A_{1500}$ by TdT polymerization

TdT polymerization is a widely employed method for synthesizing ssDNA products through controlled nucleotide polymerization. To synthesize $A_{1500}$, a mixture of 0.2 μM primer, 300 μM dATP, 1x TdT buffer, and 1 U μL$^{-1}$ TdT enzyme (Supplementary Materials) was prepared in nuclease-free water, resulting in a final volume of 150 μL. The reaction was conducted at 37 °C for 2 h and then quenched by adding 20 μL of 200 mM EDTA solution. The obtained polymer was purified using 10 kDa amicon ultra centrifugal filters and washed three times with TE buffer. The concentration of the polymer was determined using UV–Vis spectroscopy (Supplementary Characterization Methods and Instrument).

### Palindromic effect on coacervate formation

To evaluate the impact of the palindromic length and GC content on the multivalency-driven coacervation, eight different binder strands with varying palindromic lengths and sequences were introduced (Supplementary Table 1). The samples were prepared by dissolving 0.03 g L$^{-1}$ (~0.06 μM) $A_{1500}$ and 1.5 μM binder in rCutSmart buffer containing 10 mM $Mg^{2+}$ and 50 mM $K^+$ (pH = 7.9), resulting in a final volume of 20 μL. The solutions were incubated for 4 h at 37 °C with gentle rotation at 80 rpm. The

**Fig. 5 | Engineering accelerated wetting between two coacervates and its mechanisms.** Coacervates $C_1$ (or $C_3$) were formed using a mixture containing a final concentration of 0.06 μM of $A_{1500}$ and 1.5 μM of either $B_{8-1}$ or $B_{8-3}$. Coacervates $C_{31}$ were generated from 0.06 μM $A_{1500}$ and a mixture totaling 1.5 μM of $B_{8-1}$ and $B_{8-3}$. **a** Illustration of three distinct coacervate structures ($C_1$ in magenta, $C_3$ in cyan, and $C_{31}$ in dark cyan) synthesized using different binders ($B_{8-1}$, $B_{8-3}$, and a mixture of $B_{8-3}$ and $B_{8-1}$). The experiments were performed at 37 °C with gentle rotation at 80 rpm unless otherwise specified. **b** Wetting behavior upon the combination of $C_1$ and $C_3$. **c** Accelerated wetting behavior upon the combination of $C_1$ and $C_{31}$ by the introduction of $B_{8-1}$ in the $C_{31}$ system at a ratio of 75%:25% $B_{8-3}$:$B_{8-1}$ (1.125 μM of $B_{8-3}$ and 0.375 μM of $B_{8-1}$). **d** Accelerated wetting behavior upon the combination of $C_1$ and $C_{31}$ by introducing $B_{8-1}$ in the $C_{31}$ system at a ratio of 50%:50% $B_{8-3}$:$B_{8-1}$ (0.75 μM of $B_{8-3}$ and 0.75 μM of $B_{8-1}$). The experiments were conducted at 37 °C without rotation to minimize collisions. **e** Suggested mechanisms of coacervate fusion for $C_1$ and $C_3$. The molecular exchange between different-type coacervates occurs through either detachment of the binders and their diffusion or strand displacement. Scale bars: 10 μm.

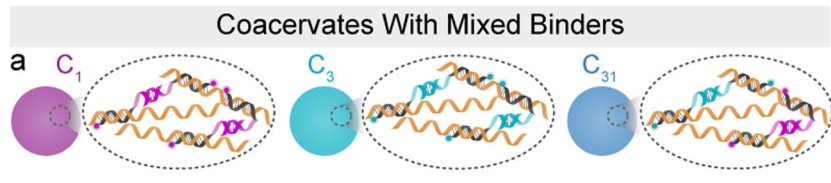

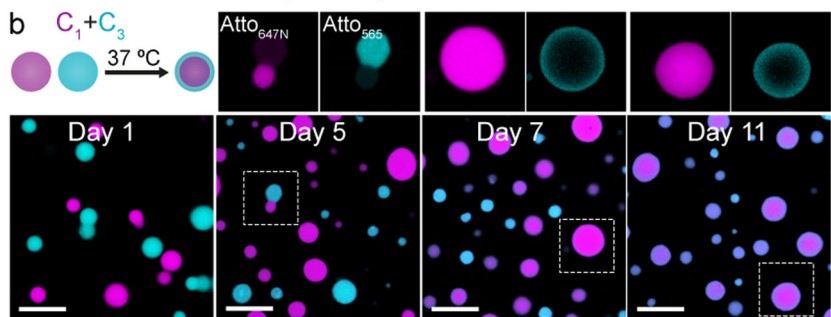

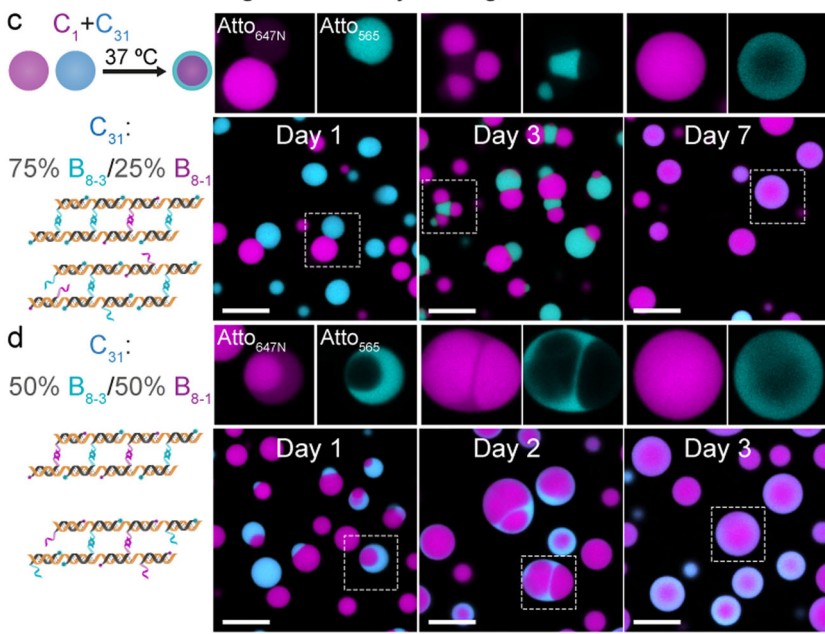

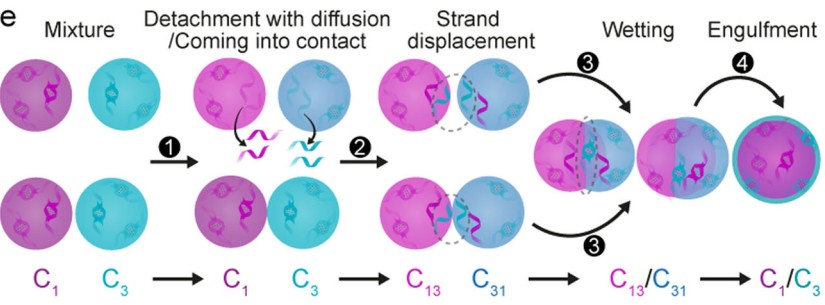

resulting structures were visualized and recorded using CLSM. The experiments were conducted using a 384-well microtiter plate with a 0.2 mm glass bottom.

### Time-dependent coacervate formation

A 20 μL solution was prepared containing 0.06 μM of $A_{1500}$ and 1.5 μM of $B_{8-1}$ in rCutSmart buffer. This solution was incubated for 30 h at 37 °C with a

rotation of 80 rpm. The coacervate formation process was recorded using CLSM.

### DNA-triggered switchable coacervate

To realize a transformation sequence from coacervate to solution and back to coacervate (coacervate-to-solution-to-coacervate), coacervates containing $B_{8-1}$ were prepared with a final concentration of 0.06 μM $A_{1500}$ and

$1.5\,\mu M$ (1.0 eq.) of $B_{8\text{-}1}$ in rCutSmart buffer. This mixture was incubated for 12 h at 37 °C with gentle rotation at 80 rpm. Subsequently, $2.25\,\mu M$ (1.5 eq.) of DNA invader was introduced and allowed to incubate for 0.5 h, resulting in the disassembly of the coacervates. Following this, $3\,\mu M$ (2.0 eq.) of DNA anti-invader was added and incubated for an additional 16 h, leading to the reformation of coacervates. Images were recorded during this process using CLSM.

### RNA-triggered switchable coacervate

In the switchable system between coacervate and solution states, coacervates containing $B_{8\text{-}1}$ were prepared with a final concentration of $0.06\,\mu M$ $A_{1500}$ and $1.5\,\mu M$ (1.0 eq.) of $B_{8\text{-}1}$ in rCutSmart buffer. This mixture was incubated for 1 h at 37 °C with gentle rotation at 80 rpm. Subsequently, RNA invader $(4.5\,\mu M,\,3\,\text{eq.})$ was introduced and allowed to incubate for 10 min until the coacervates disappeared. Following this, $0.1\,U\,\mu L^{-1}$ RNase H was added and incubated overnight.

For the transformation sequence of solution-to-coacervate-to-solution, a mixture containing $0.06\,\mu M$ $A_{1500}$, $1.5\,\mu M$ (1.0 eq.) $B_{8\text{-}1}$, and $2.25\,\mu M$ (1.5 eq.) DNA invader was prepared in rCutSmart buffer (complex of $A_{1500}$/$B_{8\text{-}1}$/DNA invader). The solution was rapidly heated to 80 °C at a rate of $3\,°C\,s^{-1}$ and then gradually cooled to 20 °C at a rate of $0.01\,°C\,s^{-1}$. Subsequently, RNA anti-invader $(15\,\mu M,\,10.0\,\text{eq.})$ was added to the annealed solution and incubated for 1 h to form coacervates. Following this, $0.1\,U\,\mu L^{-1}$ RNase H was introduced and incubated for 2 h, resulting in the coacervates disassembling into a homogeneous solution.

### RNA-regulated transient coacervate

For the RNA-regulated transient coacervate disassembly system, the coacervates were first prepared with final concentrations of $0.06\,\mu M$ of $A_{1500}$ and $1.5\,\mu M$ (1.0 eq.) of $B_{8\text{-}1}$ in rCutSmart buffer. The mixture was incubated for 2 h at 37 °C with gentle rotation at 80 rpm. Following this, varying amounts of RNA invader (15, 45, 75 μM) and $0.1\,U\,\mu L^{-1}$ RNase H were added to the solution. The mixture was then incubated at 37 °C with gentle rotation at 80 rpm. Images were recorded during this process using CLSM, and the tunable reassembly time was monitored. Additionally, to further explore the effect of RNase H concentration on the reassembly time, RNA invader $(15\,\mu M,\,10.0\,\text{eq.})$ and different RNase H concentrations $(0.03,\,0.05,\,0.1\,U\,\mu L^{-1})$ were introduced. The duration of reassembly is measured from the moment the preformed coacervates disassemble to the commencement of coacervate reformation.

To achieve the opposite pathway (transient coacervate assembly), RNA anti-invader and RNase H were introduced. The complex of $A_{1500}$/$B_{8\text{-}1}$/DNA invader was prepared with a final concentration of $0.06\,\mu M$ $A_{1500}$, $1.5\,\mu M$ (1.0 eq.) $B_{8\text{-}1}$, and $2.25\,\mu M$ (1.5 eq.) DNA invader in rCutSmart buffer. The solution was rapidly heated to 80 °C at a rate of $3\,°C\,s^{-1}$ and then gradually cooled to 20 °C at a rate of $0.01\,°C\,s^{-1}$. Then different concentrations of RNA anti-invader (15, 45 μM) and RNase H $(0.1\,U\,\mu L^{-1})$ were added to the solution. The mixture was incubated at 37 °C with gentle rotation at 80 rpm, and images were recorded during this process using CLSM.

### Engineering wetting between coacervates

Three distinct coacervate structures, denoted as $C_1$, $C_2$, and $C_3$, were prepared using different binders ($B_{8\text{-}1}$, $B_{8\text{-}2}$, and $B_{8\text{-}3}$). A mixture with a final concentration of $0.06\,\mu M$ $A_{1500}$ and $1.5\,\mu M$ of the respective $B_x$ was dissolved in rCutSmart buffer and incubated for 1 h at 37 °C with gentle rotation at 80 rpm. Subsequently, two of these resulting coacervates were mixed and incubated for 11 days at 37 °C without shaking (shaking was avoided in this case to limit collisions). The morphological changes during this period were captured and recorded using CLSM.

### Engineering wetting under nucleation process by activating $C_1$ to preformed $C_2$ (or $C_3$)

To investigate the structures that can emerge through the wetting behavior during the nucleation process, we introduced a system where $B_{8\text{-}1}$ is blocked by an RNA invader. Coacervates of $C_2$ and $C_3$ were

prepared as described previously. A mixture consisting of $A_{1500}$ $(0.06\,\mu M)$, $B_{8\text{-}1}$ $(1.5\,\mu M,\,1.0\,\text{eq.})$, and RNA invader $(2.25\,\mu M,\,1.5\,\text{eq.})$ was prepared in rCutSmart buffer, forming the complex of $A_{1500}$/$B_{8\text{-}1}$/RNA invader. The solution was rapidly heated to 80 °C at a rate of $3\,°C\,s^{-1}$ and then gradually cooled to 20 °C at a rate of $0.01\,°C\,s^{-1}$. Subsequently, $0.1\,U\,\mu L^{-1}$ RNase H and preformed $C_2$ (or $C_3$) were introduced into the solution and incubated for 72 h at 37 °C with gentle rotation at 80 rpm. The morphological changes occurring during this period were monitored and recorded using CLSM.

### Accelerated wetting between two coacervates

To facilitate the wetting process, coacervate $C_{31}$, incorporating different ratios of $B_{8\text{-}1}$ and $B_{8\text{-}3}$ ($B_{8\text{-}3}$:$B_{8\text{-}1}$ = 75%:25%, 50%:50%), was employed. In the system with 75%:25% $B_{8\text{-}3}$:$B_{8\text{-}1}$, a mixture comprising $1.125\,\mu M$ of $B_{8\text{-}3}$ and $0.375\,\mu M$ of $B_{8\text{-}1}$ was combined with $0.06\,\mu M$ $A_{1500}$ in rCutSmart buffer. This solution was then incubated for 1 h at 37 °C with gentle rotation at 80 rpm to form $C_{31}$. $C_1$ was prepared as described previously. Subsequently, $C_{31}$ solution was mixed with the $C_1$ solution and incubated for 7 days at 37 °C without shaking. Morphological changes during this period were observed and recorded using CLSM.

### Strand exchange reaction between $A_{37}$/$B_{8\text{-}1}$ and $B_{8\text{-}2}$

A mixture was prepared by mixing $2\,\mu M$ $A_{37}$ and $10\,\mu M$ $B_{8\text{-}1}$ in rCutSmart buffer, resulting in a final volume of 20 μL. This solution was rapidly heated to 80 °C at a rate of $3\,°C\,s^{-1}$ and then slowly cooled to 20 °C at a rate of $0.01\,°C\,s^{-1}$. Then $10\,\mu M$ $B_{8\text{-}2}$ was added to the solution, and the mixture was incubated for 24 h at 37 °C. The samples were analyzed through agarose gel electrophoresis using a 4 wt% agarose gel.

### Coacervate formation using commercial polyA

A 20 μL solution containing $0.06\,\mu M$ of polyA and $2.5\,\mu M$ of $B_{8\text{-}1}$ in rCutSmart buffer was prepared. This solution was incubated at 37 °C with a rotation of 80 rpm, and images were recorded during this process using CLSM.

### Reporting summary

Further information on research design is available in the Nature Portfolio Reporting Summary linked to this article.

## Data availability

The authors declare that all relevant data supporting the findings of this study are available within the paper and its Supplementary Information file. The source data for Figs. 2c and 3h-j are included in Supplementary Data 1. Additional data related to this study are available from the corresponding author upon reasonable request.

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

## Acknowledgements

This work was supported by the DFG in WA 3084/19-1. W. L. acknowledges the support by the Chinese Scholarship Council (CSC).

## Author contributions

W.L. and A.W. conceived the project, designed the experiments, analyzed the data, and wrote the manuscript. W.L. performed the experiments and collected the data. J.D. contributed to the project design and data analysis. S. Song conducted experiments for synthesizing commercial polyA-based coacervates and supported the interpretation of the results. S. Sethi assisted with the discussions of the results. A.W. supervised the work.

## Funding

## Competing interests

The authors declare no competing interests.
