## [Peer Review File · Communications Chemistry]

Reviewers' comments:

Reviewer #1 (Remarks to the Author):

The manuscript "A Facile DNA Coacervate Platform for Engineering Wetting, Engulfment, Fusion and Transient Behavior" by Liu et al reports a simple approach for the formation of coacervates using long single-stranded DNA homopolymer in combination with a series of palindromic oligonucleotide DNA binders. The authors show how the coacervates formation is regulated by sequence and length of DNA binders, and how to engineer coacervate droplets dissolution and assembly with reaction circuits based on DNA/RNA strand displacement and RNA degradation with RNase H.

Wetting and mixing of droplets in mixtures of multiple coacervate species are also reported and described based on strand exchange.

The quality of the manuscript is high and the experimental methodology is adequate. The reported results are relevant for the field.

I suggest publication of the manuscript after the following minor revisions.

Minor revisions

- 1) The dimension of the coacervate droplets is changing over time, as shown in Fig. 2e and Fig. S1. Can you please specify at what time the diameters reported in Fig. 2c have been measured?
- 2) Fig. 3h: why the disassembly time is lower for 50 eq. and 30 eq. of RNA invader with respect to 10 eq.? The contrary would be expected. Why the diameters of the initial droplets are different for different amounts of RNA invader (Fig. 3h and S4)?
- 3) The statistics of figures 3h,i,j are lacking. Can you please discuss the number of droplets analyzed and/or independent experiments performed.

Reviewer #2 (Remarks to the Author):

The manuscript describes an isothermal model system to form coacervates in vitro. The authors have introduced a ssDNA homopolymer (A1500) dispersed in the solution, which is the target for a ssDNA molecule called 'binder.' The binder is composed of four segments: a dye label, a polyT (T30) domain, an overhang domain (S7), and a programmable palindromic sticky end (Px). The T30 domain of the binder hybridizes to the homopolymers, and various binders interact through their sticky ends, leading to the formation of coacervates. By changing the Px sequence, they were able to observe a range of self-assembly regimes, from dispersed state to gel-like state. Next, they employed DNA and RNA oligos to enable strand displacement reactions (i.e., invasion and anti-invasion) to disassemble and reassemble the coacervates, offering a switchable, transient dynamic system to manipulate DNA-based droplets. They also bring into the light the dynamics of a system with multiple distinct binders over a long-term incubation. Overall, I found the idea of isothermal self-assembly of DNA droplets very compelling, and the study is suitable for publication. I have several comments and questions for improvement.

Major comments and questions:

1. I am not sure I agree with the argument/motivation that this approach of making DNA/RNA condensates is significantly simpler than assembling nanostars or using short repeats. In fact, there are previous reports of isothermal disassembly/assembly of DNA nanostars by Do et al, and Agarwal et al. Formation of DNA nanostars requires annealing, but heating followed by slow cooling is sufficient, there is no need for complex annealing protocols. I think there might be better ways to justify the approach described. Given the different nature of the binding patterns between strands, there may be rheological and mechanical properties achievable in this context that cannot be achieved through DNA nanostars.
2. Can this approach be used to build condensates that do not mix by design? This has been shown with DNA nanostars, but I don't think that is possible here because the polyA scaffold can introduce promiscuity between the binders. In fact, this is what we observe in Figs 4 and 5. If my intuition is correct, this is a disadvantage that should be discussed.
3. Please specify incubation time for each of the variants in fig. 2d. The size distribution evolves with time, so incubation time should be consistent otherwise samples can't really be compared. The SI mentions 4 hours in a rotator at 37C, is this the case for all samples? What happens in the absence of rotation?
4. Following on the previous comment, in Figure 3i and Figure S4, wherein the authors investigated disassembly and reassembly of coacervates by varying the amounts of RNA invader, why do the initial states not have similar size distributions? This could compromise the conclusion of the authors, as the initial size of the coacervates can contribute to their disassembly. For instance, in Figure S4C, it is unclear whether the slow kinetics of disassembly emanates from the high concentration of RNA invaders or it is because of the big coacervates which take longer time to dissolve.
5. What is the recovery time for 4 and 6 nt palindromic domains?
6. In the RNA/RNase H Switching, there is not sufficient information about the concentration of either molecule (Figure 3e-f), which makes it hard for a direct comparison to the results in Figure 3i-m. Perhaps in general, the authors can provide the concentration values in the captions of Figures 3 to 5 in order to consolidate the conclusions and to prevent confusion.
7. Although the images timepoints are annotated in the SI figures, the kinetics of the mechanisms in Figure 3 were not discussed thoroughly in the manuscript, e.g., is there a noticeable difference in hybridization speed of DNA invader with that of RNA invader?
8. Why were the C1/C2/C3 coacervates not incubated with a rotator after mixing, like the other samples? Does rotation accelerate the fusion dynamics? (SI section 4.7 and following)
9. For the hollow capsules due to the presence of RNase H, the reasoning and conclusion needs more discussion. Figure 3m (or Figure S6) shows a mixture of both hollow and full coacervates, and the hollow

coacervates are remarkably bigger than the full ones. This may suggest the emergence of more complex biophysical phenomena. The authors may want to refer to this paper for their discussion: <https://doi.org/10.1073/pnas.2001654117>.

10. In Figure S2b, despite the fact that the droplet fusion occurs, the FRAP result for C3 suggests there is no recovery over the course of 1.5 h. I am wondering if the plot accurately represents the images.

11. I wish there was more discussion/hypotheses as to why the binders interact the way they do in the mixed coacervates of Fig. 4 and 5. In particular: A) how do they interact? Should we assume there is some off-target binding between P8-1, P8-2, and P8-3? Can this be checked with Nupack? I think this is a reasonable assumption since the system is operating above the melting temperature for all the palindromic domains, and that the P domains were not designed to be orthogonal. Or are the polyT domains moving between polyA scaffolds? B) What is the effect of the fluorophore on the way the binders partition in the coacervates? Have the authors checked if the fluorophores are hydrophilic/hydrophobic, and how might that contribute to some binders preferring the shell vs the core of the droplets?

12. It would be helpful to quantify the results in Figure 5b-d by finding the partition coefficient of the mixed coacervates so as to compare the wetting kinetics in three mixing cases discussed in Figure 5.

13. Figure 4c (mixing of C1 and C3) and 5b are identical; this must be a mistake since for Fig. 5b the text appears to refer to mixing of C3 and C31.

Minor comments:

1. Scale bars should be reported in each microscopy image.
2. Fig 2d “effect of palindrome length” -> “effect of palindromic domain length”
3. On page 4, I suspect the word ‘approximate’ needs to be replaced by ‘average’ here: “After 0.5 h, a population of small coacervates with an approximate diameter of 0.7 μm ...”
4. The y-axes in Figure 3i-j are not consistent, causing a bit of confusion.
5. Page 5, second column: I don’t understand the meaning of “This connects to Figure 3e, where blocked building blocks were activated by RNA digestion for coacervation.”
6. In Figure S7, labels are not aligned with the lanes in the gel images.
7. In the caption of Figure 3, “(e)” needs to be “(e-f).”
8. Second to last paragraph of the discussion – what does “vs. non-dynamic fractal aggregation or systems lacking coacervation”? I don’t think the paper includes any comparison in this sense.

Reviewer #3 (Remarks to the Author):

The manuscript by Liu et al reports on a simple strategy to generate programmable DNA droplets, which may be valuable for applications in synthetic biology, smart materials, or generally to study the biophysics of biomolecular condensation. The authors use a long poly-A strands, which are cross linked by shorter staples with a 30T domain and a palindromic sticky end (and a toehold region between the two). Condensate formation is readily observed, with trends noted as a function of sticky-end length that

fit with expectations. Triggered assembly and disassembly can be achieved by blocking and unblocking the palindromic domains using DNA/RNA oligonucleotides, strand displacement reactions and enzymatic degradation of RNA using RNase-H. Finally, the authors demonstrate interesting mixing behaviours when combining pre-formed condensates with different palindromic sticky ends, triggered by the slow exchange between the staple strands.

Overall, this is an excellent manuscript that nicely shows how robust condensate formation and programmable dynamics can be achieved with simple building blocks. The insights in the mixing dynamics are also interesting. The manuscript is clear and easy to read. I would recommend publication in *Communications Chemistry* once the authors have addressed the minor points below.

1) Could the authors clarify how the reassembly times in Fig. 3i and h were extracted from the diameter vs time curves?

2) With respect to the experiment in Fig. 3l and m, the authors ascribe the formation of hollow condensates to trapping of the RNase H within the cavities. Can the authors expand on this interpretation? Why would the polymerase remain trapped? And why would the cavities formed not diffuse through the condensates and be “released” on its surface?

3) Panels c in Fig. 3 and b in Fig. 4 are the same. I don't think there is need to show the same panel twice and it is preferable to refer to 3b when discussing Fig. 4.

4) Can the authors discuss more in detail their hypothesis for the mechanism leading to progressive mixing of distinct condensate types? Do they expect the initial exchange of staples (B8-1, B8-2 etc) between different condensates to occur through complete detachment of the staples and their diffusion through the bulk or only via proximity-induced strand displacement during brief contacts between different-type condensates? To this end, Fig. 5e could be clearer (in the initial step the condensates are shown as close to each other, but are they in contact?)

5) In the experiments shown in Fig 5c and d, do the percentages of the staples reported for the mixed C31 condensates refer to the overall binding capacity of the polyA strands or to the total amount of staples used for prior experiments (50% binding capacity)? In other words, are polyA strands in C31 condensates nominally fully saturated with staples?

6) In page 4, right column, 3rd paragraph, the authors say “A toehold is no longer needed as for the RNA invader”. I believe this should say “DNA invader”.

Reviewer #4 (Remarks to the Author):

The manuscript by Liu et al. details the design and construction of DNA-based coacervates and explores various strategies for controlling their reversible formation and dissolution. These coacervates are constructed from long single-stranded polyA combined with short oligonucleotides “crosslinkers”, which bind to polyA and each other via a palindromic sequence. Depending on the sequence of these binders, more or less fluidic structures are produced. The binders are also designed to respond to the presence of

invaders oligos or enzyme-mediated degradation. This design is leveraged to create switchable coacervates that form and dissolve reversibly upon the introduction of a cue. The kinetic modulation of two antagonistic reactions (hybridization vs. enzymatic degradation) last achieves transient coacervate dissolution and formation, showcasing autonomous behavior.

There is currently a growing interest in designing dynamic and programmable coacervates, and DNA nanotechnology offers a particularly promising tool in this pursuit. In contrast to previously reported DNA coacervates typically assembled from networks of DNA nanostars, the authors' approach offers a simpler, versatile and elegant method. The manuscript is clear and concise and data well presented. For these reasons, I support publication in Communication Chemistry. I have a few minor comments below.

1/ On page 4 about the FRAP results and fusion of droplets: it appears that FRAP reveals very limited fluorescence recovery, indicating slow diffusion of DNA strands. Similarly, the fusion events do not seem to fully complete within the experimental timescale, as relaxation to a spherical droplet is not achieved. Thus, I suggest rephrasing this paragraph to emphasize the limited FRAP recovery and only partial fusion.

2/ On page 5, concerning the vacuoles formed during enzyme-driven coacervate dissolution: Similar vacuolization during coacervate dissolution has been observed and studied previously. The authors should reference the work of Saleh, Jeong, Liedl, PNAS, 2020, 117, 16160-16166 here.

3/ Figure 3f: Small objects are visible with the DNA invader (prior to the addition of RNA anti-invader). Could this be due to the DNA invader's incomplete dissolution of the droplets? Please comment.

4/ Methods 4.7 "engineering wetting between coacervates": Was the incubation for 11 days performed in test tubes or directly in a microscope observation chamber? If conducted in test tubes, was sedimentation observed?

5/ The authors mention the potential formulation of their systems using commercially available polyA. To further demonstrate the general applicability of these coacervates, it would be interesting to showcase an example using commercial polyA.

Response Letter to Reviewer's Comments

Reviewer #1

The manuscript "A Facile DNA Coacervate Platform for Engineering Wetting, Engulfment, Fusion and Transient Behavior" by Liu et al reports a simple approach for the formation of coacervates using long single-stranded DNA homopolymer in combination with a series of palindromic oligonucleotide DNA binders. The authors show how the coacervates formation is regulated by sequence and length of DNA binders, and how to engineer coacervate droplets dissolution and assembly with reaction circuits based on DNA/RNA strand displacement and RNA degradation with RNase H.

Wetting and mixing of droplets in mixtures of multiple coacervate species are also reported and described based on strand exchange.

The quality of the manuscript is high and the experimental methodology is adequate. The reported results are relevant for the field.

I suggest publication of the manuscript after the following minor revisions..

Response: We thank the reviewer for their supportive and encouraging comment.

Minor revisions:

1. The dimension of the coacervate droplets is changing over time, as shown in Fig. 2e and Fig. S1. Can you please specify at what time the diameters reported in Fig. 2c have been measured?

Response: The data in Fig. 2c were measured after a 4 h. We specified this in the caption, including other details of the system.

2. Fig. 3h: why the disassembly time is lower for 50 eq. and 30 eq. of RNA invader with respect to 10 eq.? The contrary would be expected. Why the diameters of the initial droplets are different for different amounts of RNA invader (Fig. 3h and S4).

Response: We believe there may be some misunderstanding. The disassembly times for all the conditions, including those with 10 eq., 30 eq., and 50 eq. of RNA invader, are actually nearly identical. The missing data at 10 minutes for the systems with 30 eq. and 50 eq. of RNA invader have now been included in Figure 3h.

The differing diameters of the initial droplets resulted from the broad diameter distribution of the coacervates. For clarity, we calculated the average diameter and replotted the data in Figure 3h and 3i.

3. The statistics of figures 3h,i,j are lacking. Can you please discuss the number of droplets analyzed and/or independent experiments performed.

Response: The statistics have been added to the caption of Figure 3h-j in MS (Page 5) and additional details of corresponding experiments have been added in the Methods section of MS (Page 8).

Reviewer #2

The manuscript describes an isothermal model system to form coacervates in vitro. The authors have introduced a ssDNA homopolymer (A_{1500}) dispersed in the solution, which is the target for a ssDNA molecule called 'binder.' The binder is composed of four segments: a dye label, a polyT (T_{30}) domain, an overhang domain (S_7), and a programmable palindromic sticky end (P_x). The T_{30} domain of the binder hybridizes to the homopolymers, and various binders interact through their sticky ends, leading to the formation of coacervates. By changing the P_x sequence, they were able to observe a range of self-assembly regimes, from dispersed state to gel-like state. Next, they employed DNA and RNA oligos to enable strand displacement reactions (i.e., invasion and anti-invasion) to disassemble and reassemble the coacervates, offering a switchable, transient dynamic system to manipulate DNA-based droplets. They also bring into the light the dynamics of a system with multiple distinct binders over a long-term incubation. Overall, I found the idea of isothermal self-assembly of DNA droplets very compelling, and the study is suitable for publication. I have several comments and questions for improvement.

Response: We thank the reviewer for their support and encouragements.

Major comments and questions:

1. I am not sure I agree with the argument/motivation that this approach of making DNA/RNA condensates is significantly simpler than assembling nanostars or using short repeats. In fact, there are previous reports of isothermal disassembly/assembly of DNA nanostars by Do et al, and Agarwal et al. Formation of DNA nanostars requires annealing, but heating followed by slow cooling is sufficient, there is no need for complex annealing protocols. I think there might be better ways to justify the approach described. Given the different nature of the binding patterns between strands, there may be rheological and mechanical properties achievable in this context that cannot be achieved through DNA nanostars.

Response: We thank the reviewer for this comment. The coacervate system using nanostars requires at least several different DNA strands to form specific nanostar structures. Additionally arm flexibility needs to be controlled and detailed knowledge on this is necessary. Furthermore, as highlighted by Agarwal *et al.*, an annealing process for the formation of the DNA nanostars is essential before initiating isothermal condensation.¹ We also note that other reviewers (e.g. Reviewer#4) agree with the notion on simplicity.

Our approach is indeed a bit simpler to nanostars as this knowledge is not needed. Our approach focuses primarily on a short strand – the design of the binder. This streamlined design eliminates the need for annealing and only requires incubation at 37 °C. Therefore, in terms of both design and procedural simplicity, our system offers a more straightforward approach compared to the assembly of nanostars. We however agree to the notion of different viscoelastic properties and added this to the MS.

2. Can this approach be used to build condensates that do not mix by design? This has been shown with DNA nanostars, but I don't think that is possible here because the polyA scaffold can introduce promiscuity between the binders. In fact, this is what we observe in Figs 4 and 5. If my intuition is correct, this is a disadvantage that should be discussed.

Response: We now commented on this in the conclusion section that longer T_x segments that bind the palindromic linkers on the polyA scaffold are likely able to prevent fusion (Page 7). It however requires a detailed follow up study to understand quantitatively how the length of the T_x sticker suppresses mixing of condensates. Here the focus is on the mixing as detailed in Fig. 4 and 5.

3. Please specify incubation time for each of the variants in fig. 2d. The size distribution evolves with time, so incubation time should be consistent otherwise samples can't really be compared. The SI mentions 4 hours in a rotator at 37C, is this the case for all samples? What happens in the absence of rotation.

Response: A 4-hour incubation period at 37 °C with rotation was employed for all samples in Figure 2d. Details now in caption. In absence of rotation, the kinetics of coacervation seems to be slower, but we did not quantify these details to the extent that we want to discuss them in this article.

4. Following on the previous comment, in Figure 3i and Figure S4, wherein the authors investigated disassembly and reassembly of coacervates by varying the amounts of RNA invader, why do the initial states not have similar size distributions? This could compromise the conclusion of the authors, as the initial size of the coacervates can contribute to their disassembly. For instance, in Figure S4C, it is unclear whether the slow kinetics of disassembly emanates from the high concentration of RNA invaders or it is because of the big coacervates which take longer time to dissolve.

Response: This is similar to comment 2 by Reviewer #1. We believe there may be some misunderstanding. The disassembly times for all the conditions, including those with 10 eq., 30 eq., and 50 eq. of RNA invader, are actually nearly identical. The missing data at 10 minutes for the systems with 30 eq. and 50 eq. of RNA invader have now been included in Figure 3h.

The differing diameters of the initial droplets resulted from the broad diameter distribution of the coacervates. For clarity, we calculated the average diameter and replotted the data in Figure 3h and 3i.

5. What is the recovery time for 4 and 6 nt palindromic domains.

Response: It is slightly faster compared to 8 nt. Overall, however, not significantly different. We added this data to new Figure S3, and made a comment in the FRAP section on page 4.

6. In the RNA/RNase H Switching, there is not sufficient information about the concentration of either molecule (Figure 3e-f), which makes it hard for a direct comparison to the results in Figure 3i-m. Perhaps in general, the authors can provide the concentration values in the captions of Figures 3 to 5 in order to consolidate the conclusions and to prevent confusion.

Response: The concentration values have been added in the captions of Figures 3 to 5.

7. Although the images timepoints are annotated in the SI figures, the kinetics of the mechanisms in Figure 3 were not discussed thoroughly in the manuscript, e.g., is there a noticeable difference in hybridization speed of DNA invader with that of RNA invader.

Response: When DNA or RNA invaders were introduced, the coacervates disassembled rapidly, in less than 10 minutes. The speed at which DNA or RNA invaders hybridized with DNA binders showed some slight differences (for molecular systems: earlier studies by Liu *et.al* and Bishop *et.al.*^{2,3})

The focus of this MS is however not to understand minor differences in DNA vs RNA hybridization, but to show the behavior on a systems level. It is more important to understand e.g. influence of the RNase concentration or influence of RNA concentration. We therefore prefer to not comment on these differences. We believe that polydisperse condensates are not ideal to unravel such details with highest level of quantification. We added similar disassembly times to the respective text parts.

8. Why were the C1/C2/C3 coacervates not incubated with a rotator after mixing, like the other samples? Does rotation accelerate the fusion dynamics? (SI section 4.7 and following).

Response: Yes, we wanted to reduce impact of collision to due convection.

9. For the hollow capsules due to the presence of RNase H, the reasoning and conclusion needs more discussion. Figure 3m (or Figure S6) shows a mixture of both hollow and full coacervates, and the hollow coacervates are remarkably bigger than the full ones. This may suggest the emergence of more complex biophysical phenomena. The authors may want to refer to this paper for their discussion: <https://doi.org/10.1073/pnas.2001654117>.

Response: We thank reviewer for their valuable suggestion. We have added some discussion and the reference (Page 5).⁴ However, mechanistically both systems are very different as the vacuole formation in the DNA nanostar system was achieved by adding the enzyme from the outside after condensate formation, whereas here entrapment occurs.

10. In Figure S2b, despite the fact that the droplet fusion occurs, the FRAP result for C3 suggests there is no recovery over the course of 1.5 h. I am wondering if the plot accurately represents the images.

Response: There is some recovery, even though limited. The data is accurate.

11. I wish there was more discussion/hypotheses as to why the binders interact the way they do in the mixed coacervates of Fig. 4 and 5. In particular: A) how do they interact? Should we assume there is some off-target binding between P8-1, P8-2, and P8-3? Can this be checked with Nupack? I think this is a reasonable assumption since the system is operating above the melting temperature for all the palindromic domains, and that the P domains were not designed to be orthogonal. Or are the polyT domains moving between polyA scaffolds? B) What is the effect of the fluorophore on the way the binders partition in the coacervates? Have the authors checked if the fluorophores are hydrophilic/hydrophobic, and how might that contribute to some binders preferring the shell vs the core of the droplets?

Response: A) There is no indication for off-target binding for B₈₋₁, B₈₋₂, and B₈₋₃, as well as for P₈₋₁, P₈₋₂, and P₈₋₃, at 37 °C or even at 5 °C, as confirmed by Nupack analysis. We added a comment on page 6 in MS.

B) We agree with the reviewer regarding the potential impact of the hydrophilicity of the fluorophores on the wetting behavior. We included a statement of the hydrophilicity of the dye. Atto₄₈₈ is the most hydrophilic dye, but it is not the material that always ends up on the surface during mixing. This warrants follow up investigations to really understand this fully. But that requires significant modulation of all components, which is beyond this article introducing the coacervate platform.

12. It would be helpful to quantify the results in Figure 5b-d by finding the partition coefficient of the mixed coacervates so as to compare the wetting kinetics in three mixing cases discussed in Figure 5.

Response: We did not understand this question. The coacervates are in parts premixed and if there is any exchange it would occur at the interface, which means it is ambiguous to determine any change in partition coefficient as this has a spatial aspect to it. At which spatial position would one calculate this partition?

13. Figure 4c (mixing of C₁ and C₃) and 5b are identical; this must be a mistake since for Fig. 5b the text appears to refer to mixing of C₃ and C₃₁.

Response: We believe there may be some misunderstandings. In the manuscript, we wrote: "Upon mixing C₁ and C₃₁, core-shell structures visually identical to the C₁/C₃ system (Figure 5b) are formed". To clarify and enhance understanding, we have changed this to: "Upon mixing C₁ and C₃₁ (Figure 5c,d), core-shell structures are formed that are visually identical to those the C₁/C₃ system (Figure 4c and Figure 5b)" (Page 7). It is correct that we included data from Figure 4c into figure 5 for direct comparison. We believe this benefits the reader.

Minor comments:

1. Scale bars should be reported in each microscopy image.

Response: Scale bars have been added in each microscopy image.

2. Fig 2d “effect of palindrome length” -> “effect of palindromic domain length”.

Response: We corrected “effect of palindrome length” to “effect of palindromic domain length”.

3. On page 4, I suspect the word ‘approximate’ needs to be replaced by ‘average’ here: “After 0.5 h, a population of small coacervates with an approximate diameter of 0.7 μm ...”

Response: The word “approximate” has been replaced by “average”.

4. The y-axes in Figure 3i-j are not consistent, causing a bit of confusion.

Response: We have modified the y-axes in Figure 3j to ensure consistency with the y-axes in Figure 3i.

5. Page 5, second column: I don’t understand the meaning of “This connects to Figure 3e, where blocked building blocks were activated by RNA digestion for coacervation.”

Response: The sentence has been changed. We just connect to results obtained earlier.

6. In Figure S7, labels are not aligned with the lanes in the gel images.

Response: The labels have been reorganized.

7. In the caption of Figure 3, “(e)” needs to be “(e-f).”

Response: The caption has been corrected.

8. Second to last paragraph of the discussion – what does “vs. non-dynamic fractal aggregation or systems lacking coacervation”? I don’t think the paper includes any comparison in this sense.

Response: We described the effect of palindromic domain length and palindromic sequence on the morphology formation, including dynamic spherical coacervates, non-dynamic fractal aggregation, and systems lacking coacervation. Therefore, we mentioned this in the manuscript: “Using different palindromic sequences, we described the engineering space for how to obtain dynamic and spherical coacervates vs. non-dynamic fractal aggregation or systems lacking coacervation”. This is correct.

Reviewer #3

The manuscript by Liu et al reports on a simple strategy to generate programmable DNA droplets, which may be valuable for applications in synthetic biology, smart materials, or generally to study the biophysics of biomolecular condensation. The authors use a long poly-A strands, which are cross linked by shorter staples with a 30T domain and a palindromic sticky end (and a toehold region between the two). Condensate formation is readily observed, with trends noted as a function of sticky-end length that fit with expectations. Triggered assembly and disassembly can be achieved by blocking and unblocking the palindromic domains using DNA/RNA oligonucleotides, strand displacement reactions and enzymatic degradation of RNA using RNase-H. Finally, the authors demonstrate interesting mixing behaviours when combining pre-formed condensates with different palindromic sticky ends, triggered by the slow exchange between the staple strands.

Overall, this is an excellent manuscript that nicely shows how robust condensate formation and programmable dynamics can be achieved with simple building blocks. The insights in the mixing dynamics are also interesting. The manuscript is clear and easy to read. I would recommend publication in Communications Chemistry once the authors have addressed the minor points below.

Response: We thank the reviewer for their supportive and encouraging comment.

1. Could the authors clarify how the reassembly times in Fig. 3i and h were extracted from the diameter vs time curves?

Response: This methodology for determining the reassembly time has been added in the MS and Methods section.

2. With respect to the experiment in Fig. 3l and m, the authors ascribe the formation of hollow condensates to trapping of the RNase H within the cavities. Can the authors expand on this interpretation? Why would the polymerase remain trapped? And why would the cavities formed not diffuse through the condensates and be “released” on its surface.

Response: DNA/RNA enzymes stick to their targets and also off-target nucleic acids. We believe this is sufficiently clear. The droplets are viscoelastic in nature with slow recovery according to FRAP. Strictly speaking we cannot exclude whether some cavities diffuse to the surface. But this requires a detailed study, only focusing on mechanistic details. We added some more text on page 5.

3. Panels c in Fig. 3 and b in Fig. 4 are the same. I don't think there is need to show the same panel twice and it is preferable to refer to 3b when discussing Fig. 4.

Response: We believe the reviewer is indicating that Fig. 4c and Fig. 5b are identical. While we understand this is redundant, we believe this presentation aids very much in a clearer and more immediate understanding of the wetting behaviors. We now reference both figures to avoid any confusion.

4. Can the authors discuss more in detail their hypothesis for the mechanism leading to progressive mixing of distinct condensate types? Do they expect the initial exchange of staples (B8-1, B8-2 etc) between different condensates to occur through complete detachment of the staples and their diffusion through the bulk or only via proximity-induced strand displacement during brief contacts between different-type condensates? To this end, Fig. 5e could be clearer (in the initial step the condensates are shown as close to each other, but are they in contact?)

Response: We hypothesize that both mechanism could occur. We have added two sentences in the text to reflect both proposed mechanisms (Page 7). We also have changed the Fig. 5e.

5. In the experiments shown in Fig 5c and d, do the percentages of the staples reported for the mixed C31 condensates refer to the overall binding capacity of the polyA strands or to the total amount of staples used for prior experiments (50% binding capacity)? In other words, are polyA strands in C31 condensates nominally fully saturated with staples?

Response: In C_{31} coacervates, approximately 50% of the binding capacity is utilized for staples. This is consistent with the binding ratios designed in our prior experiments. The specific DNA concentration information can be found in the caption of Figure 5c,d and the Methods section of MS.

6. In page 4, right column, 3rd paragraph, the authors say “A toehold is no longer needed as for the RNA invader”. I believe this should say “DNA invader”.

Response: We thank the reviewer for this comment and the term “RNA invader” has been corrected to “DNA invader”.

Reviewer #4

The manuscript by Liu et al. details the design and construction of DNA-based coacervates and explores various strategies for controlling their reversible formation and dissolution. These coacervates are constructed from long single-stranded polyA combined with short oligonucleotides “crosslinkers”, which bind to polyA and each other via a palindromic sequence. Depending on the sequence of these binders, more or less fluidic structures are produced. The binders are also designed to respond to the presence of invaders oligos or enzyme-mediated degradation. This design is leveraged to create switchable coacervates that form and dissolve reversibly upon the introduction of a cue. The kinetic modulation of two antagonistic reactions (hybridization vs. enzymatic degradation) last achieves transient coacervate dissolution and formation, showcasing autonomous behavior.

There is currently a growing interest in designing dynamic and programmable coacervates, and DNA nanotechnology offers a particularly promising tool in this pursuit. In contrast to previously reported DNA coacervates typically assembled from networks of DNA nanostars, the authors' approach offers a simpler, versatile and elegant method. The manuscript is clear and concise and data well presented. For these reasons, I support publication in Communication Chemistry. I have a few minor comments below.

Response: We thank the reviewer for their support and encouragements.

1. On page 4 about the FRAP results and fusion of droplets: it appears that FRAP reveals very limited fluorescence recovery, indicating slow diffusion of DNA strands. Similarly, the fusion events do not seem to fully complete within the experimental timescale, as relaxation to a spherical droplet is not achieved. Thus, I suggest rephrasing this paragraph to emphasize the limited FRAP recovery and only partial fusion.

Response: We thank reviewer for the valuable comment. We have rephrased the paragraph stating the limited FRAP recovery and the partial fusion.

2. On page 5, concerning the vacuoles formed during enzyme-driven coacervate dissolution: Similar vacuolization during coacervate dissolution has been observed and studied previously. The authors should reference the work of Saleh, Jeong, Liedl, PNAS, 2020, 117, 16160-16166 here.

Response: We thank reviewer for the valuable suggestion. We have added some discussion and the reference (Page 5).⁴ However, mechanistically both systems are very different as the vacuole formation in the DNA nanostar system was achieved by adding the enzyme from the outside after condensate formation, whereas here it is present from the beginning and entrapped due to the affinity to the nucleic acids.

3. Figure 3f: Small objects are visible with the DNA invader (prior to the addition of RNA anti-invader). Could this be due to the DNA invader's incomplete dissolution of the droplets? Please comment.

Response: One sentence has been added to the caption of Figure 3f.

4. Methods 4.7 “engineering wetting between coacervates”: Was the incubation for 11 days performed in test tubes or directly in a microscope observation chamber? If conducted in test tubes, was sedimentation observed?

Response: The incubation was performed directly in a 384-well microtiter plate with a 0.2 mm black glass bottom. We used a TOKAT HIT Stage Top incubator equipped with a microscope observation chamber.

5. The authors mention the potential formulation of their systems using commercially available polyA. To further demonstrate the general applicability of these coacervates, it would be interesting to showcase an example using commercial polyA.

Response: We appreciate the suggestion. We acquired commercial polyA and formed condensates. This highlights the general applicability of our DNA coacervate system. This data is now in Figure S9 and has been discussed in the MS.

References

1. Agarwal, S., Osmanovic, D., Klocke, M.A. & Franco, E. The Growth Rate of DNA Condensate Droplets Increases with the Size of Participating Subunits. *ACS Nano* **16**, 11842-11851 (2022).
2. Liu, H. et al. Kinetics of RNA and RNA:DNA Hybrid Strand Displacement. *ACS Synth. Biol.* **10**, 3066-3073 (2021).
3. Bishop, J.O. Molecular Hybridization of Ribonucleic Acid with a Large Excess of Deoxyribonucleic Acid. *Biochem. J.* **126**, 171-185 (1972).
4. Saleh, O.A., Jeon, B.J. & Liedl, T. Enzymatic degradation of liquid droplets of DNA is modulated near the phase boundary. *PNAS* **117**, 16160-16166 (2020).

REVIEWERS' COMMENTS:

Reviewer #1 (Remarks to the Author):

The authors answered to my revision requests. I suggest publication of the manuscript.

Reviewer #2 (Remarks to the Author):

The authors have addressed most of my comments. I have a few follow ups:

Response to my major comment n. 1:

In response to my suggestion that these coacervates could have different behavior when compared to DNA-nanostar condensates, the discussion was modified to say that “different viscoelastic regime can be generated within the coacervates as compared to nanostar-based droplets” - But what kind of differences?

Response to my major comments 3 and 8: I think you should acknowledge prominently the different formation protocol adopted for the assembly/disassembly experiments, which were achieved using a rotator to accelerate strand exchange/droplet collisions, and for the multi-layer condensates, which were done w/o a rotator to reduce exchange/collisions.

Someone attempting to replicate your work would fail if they missed this important detail.

Response to my major comment n. 4, in regards to Fig 3h:

The data corresponding to the initial size distributions have actually been removed when compared to the original version of figure 3h.

In the original version the disassembly times were very different:

The 10eq sample disassembled in 15 min, while 30eq and 50 eq in about 30 min.

But the initial diameter of the 10eq sample was also about half the initial diameter of 30 and 50eq.

My question was: why is the initial size distribution so different, between samples, and could there be an inverse correlation of initial size with the disassembly time?

Response to my major comment 12: I agree, this might be too hard.

Finally, I suggest including the following new references

Maruyama T, Gong J, Takinoue M. Temporally controlled multistep division of DNA droplets for dynamic artificial cells. ChemRxiv. 2024; doi:10.26434/chemrxiv-2024-z67br

Agarwal S, Osmanovic D, Dizani M, Klocke MA, Franco E. Dynamic control of DNA condensation. Nature Communications. 2024 Mar 1;15(1):1915.

Skipper K, Wickham S. Core-shell coacervates formed from DNA nanostars. ChemRxiv. 2024;
doi:10.26434/chemrxiv-2024-07pvd

Reviewer #3 (Remarks to the Author):

I have reviewed the response and changes applied by the authors and found that they address well my minor concerns. I am thus happy to recommend publication.

Reviewer #4 (Remarks to the Author):

In their revised manuscript, the authors have addressed all my concerns. I now fully support publication in Communications Chemistry.

Response Letter to Reviewer's Comments

Reviewer #1 (Remarks to the Author):

The authors answered to my revision requests. I suggest publication of the manuscript.

Response: We thank the reviewer for the supportive comment.

Reviewer #2 (Remarks to the Author):

The authors have addressed most of my comments. I have a few follow ups:

1. Response to my major comment n. 1:

In response to my suggestion that these coacervates could have different behavior when compared to DNA-nanostar condensates, the discussion was modified to say that “different viscoelastic regime can be generated within the coacervates as compared to nanostar-based droplets” - But what kind of differences?

Response: We added: ..”because polymer entanglements can be present for long polyA, as well as higher multivalency effect. FRAP measurements indicate a slow and partial recovery in the coacervates, contrasting with the rapid and almost complete recovery reported in nanostar-based droplets by previous studies.22, 43-45”

2. Response to my major comments 3 and 8:

I think you should acknowledge prominently the different formation protocol adopted for the assembly/disassembly experiments, which were achieved using a rotator to accelerate strand exchange/droplet collisions, and for the multi-layer condensates, which were done w/o a rotator to reduce exchange/collisions.

Someone attempting to replicate your work would fail if they missed this important detail.

Response: The formation protocols have been added to the captions of Figures 2 to 5 in MS.

3. Response to my major comment n. 4, in regards to Fig 3h:

The data corresponding to the initial size distributions have actually been removed when compared to the original version of figure 3h.

In the original version the disassembly times were very different:

The 10eq sample disassembled in 15 min, while 30eq and 50 eq in about 30 min.

But the initial diameter of the 10eq sample was also about half the initial diameter of 30 and 50eq.

My question was: why is the initial size distribution so different, between samples, and could there be an inverse correlation of initial size with the disassembly time?

Response: The differing diameters of the initial coacervates are in the margin of error given the wide diameter distribution of the coacervates. Everything beyond would be speculative.

4. Response to my major comment 12: I agree, this might be too hard.

Response: We thank the reviewer for the supportive comment.

5. Finally, I suggest including the following new references

Maruyama T, Gong J, Takinoue M. Temporally controlled multistep division of DNA droplets for dynamic artificial cells. ChemRxiv. 2024; doi:10.26434/chemrxiv-2024-z67br

Agarwal S, Osmanovic D, Dizani M, Klocke MA, Franco E. Dynamic control of DNA condensation. Nature Communications. 2024 Mar 1;15(1):1915.

Skipper K, Wickham S. Core-shell coacervates formed from DNA nanostars. ChemRxiv. 2024; doi:10.26434/chemrxiv-2024-07pvd

Response: The references have been cited in the Conclusions section in MS.

Reviewer #3 (Remarks to the Author):

I have reviewed the response and changes applied by the authors and found that they address well my minor concerns. I am thus happy to recommend publication.

Response: We thank the reviewer for the supportive comment.

Reviewer #4 (Remarks to the Author):

In their revised manuscript, the authors have addressed all my concerns. I now fully support publication in Communications Chemistry.

Response: We thank the reviewer for their supportive comment.